# Decomposition analysis of women's empowerment-based inequalities in the use of maternal health care services in Ethiopia: Evidence from Demographic and Health Surveys

**Gebretsadik Shibre**[1]*, **Wubegzier Mekonnen**[1], **Damen Haile Mariam**[2]

**1** Department of Reproductive, Family and Population Health, School of Public Health, Addis Ababa University, Addis Ababa, Ethiopia, **2** Department of Health system and Policy, School of Public Health, Addis Ababa University, Addis Ababa, Ethiopia

* gebretsh@gmail.com

## Abstract

### Background

The use of maternal health care services tends to rise with women's empowerment. However, disparities in the use of maternal health care services in Ethiopia that are founded on women's empowerment are not sufficiently addressed. In light of women's empowerment equity stratifier, this study seeks to assess inequalities in the uptake of maternal health care services (early antenatal care, four or more antenatal care and postnatal care services).

### Methods

Drawing on data from the four rounds of Ethiopia Demographic and Health Surveys (EDHSs) conducted between 2000 and 2016, we conducted analysis of inequalities in utilization of maternal health care services using women's empowerment as equity stratifier. We utilized concentration index and concentration curve for assessing the inequalities. We used *clorenz and conindex* Stata modules to compute the index and curve. Decomposition of the Erreygers normalized concentration index was done to explain the inequalities in terms of other variables' percent contributions. Complex aspect of the EDHSs data was considered during analysis to produce findings consistent with the data generating process. All analyses were done using Stata v16.

### Results

Utilization of maternal health care services was inequitably distributed between empowered and poorly empowered women, with women in the highly empowered category taking more of the services. For instance, the Erreygers index for quality ANC are 0.240 (95% CI 0.207, 0.273); 0.20 (95% CI 0.169, 0.231) and 0.122 (95% CI 0.087, 0.157), respectively, for the attitude towards violence, social independence and decision-making domains of women's

**Data Availability Statement:** The women's recode file (IR file), which is accessible via the DHS website (http://www.dhsprogram.com), was used

for this study. After logging into the DHS program website, independent researchers can access the data and submit a brief justification for the release of the data that includes an overview of their intended study.

**Funding:** The research has received funding from Addis Ababa University. The funder had no role in study design, data collection and analysis, decision to publish, or preparation of the manuscript.

**Competing interests:** The authors have declared that no competing interests exist.

empowerment. Inequalities in the distribution of other variables like wealth, education, place of residence and women's empowerment itself underpin the inequalities in the utilization of the services across the women's empowerment groups.

## Conclusions

Equity in maternal health care services can be improved through redistributive policies that attempt to fairly distribute the socioeconomic determinants of health such as wealth and education between highly and poorly empowered women.

## Introduction

Maternal health care services are essential component of health care services provided to women during and after pregnancy. In addition to sustaining and enhancing women's health and well-being after giving birth, maternal health care services seek to ensure that pregnant women are in the greatest possible health [1, 2].

Worldwide, the coverage of maternal health care services has improved over time. Between 2015–2021, 45.3%, 61.2% and 91%, respectively, of women received at least four ANC service in low-income, lower middle-income and upper middle countries, with the utilization being differed greatly by whether or not women live in rural vs urban settings and in poorest vs richest households. Not surprisingly, receipt of ANC quality lags noticeably behind at least once ANC coverage (measured merely on ANC visit), with low-income countries having the highest ANC coverage-quality gap. Similarly, the prevalence of postnatal care within 2 days of birth is 49.4% and 70% in low income and lower-middle income countries, respectively, again with unequal utilization of the service across the different categories of place of residence and wealth status of households [3, 4].

However, important disparities were also witnessed in the coverage of the services within and between continents. In Sub-Saharan Africa (SSA), nearly 54% of women received at least four ANC visits, and urban dwellers had 20 percentage points (pp) higher than women in rural setting. ANC service use tends to substantially vary by whether women resided in poorest or wealthiest households, where only 39% women in poorest households received the service compared to 74% women in better-off households. Similarly, in South Asia, about 55% of women utilized ANC at least four times, with noticeable area and wealth-based inequalities. While it is 66.8% in urban areas, only 49.8% of women were able to receive the service in rural areas. In terms of wealth related gap around the service in the region, only 24% of women in poorest households got the service compared to 74% in richest wealth quintile, resulting in a difference of 50 pp. Receipt of at least four ANC is higher in Latin America and the Caribbean (LAC) and in East Asia and Pacific region compared with that of SSA and South Asia. In LAC, more than 90% of women got the service, with rural having a coverage of 86%. In East Asia and the Pacific region, 87.7% of women got ANC at least four times, with rural area having 82.1% [3].

Similarly, utilization of postnatal care service within 2 days of birth differs by various background characteristics of women, including residence type. In South Asia, 74% of women received postnatal care (PNC) within 2 days after birth between 2015–2021, with urban areas having ANC coverage more than 11 percentage points higher than in rural areas. Women in poorest households have 32 pp lower coverage of the service than women in richest households. In SSA, however, utilization of the PNC is by far lower and the extent of its variation is

more pronounced; where only 50.6% of women received the service. The urban-rural gap around the use of the service in the region is approximately 22 percentage points, suggesting urban women's greater chance of getting the service. Women who live in poorest households had 37 pp lower coverage of the service than women in the richest households.

Deeper analysis of a country's situation reveals that such aggregate statistics has hidden unacceptably noticeable gaps between different groups of women within a country. The socio-economic status and geographic location of the women would likely influence how often they use the services [4–7].

Our survey of the literature revealed that studies were done on inequalities of maternal health care services utilization using different equity stratifiers such as wealth, education as well as place where women live in [4, 8–13]. There are no inequality studies, however, on maternal health care use by women's empowerment equity stratifier. The World Health Organization (WHO) advises performing equity analysis of health care services using any equity stratifiers that are relevant for a particular health indicator of interest [14] for the purpose of monitoring the global attainment of the Sustainable Development Goals (SDGs) [15]. Goals 3 and 5, and the associated targets 3.8 and 5.6, of the SDGs call for universal coverage of sexual, reproductive health, reproductive rights and maternal health services by 2030. These goals would be attained only when drivers of health services uptake are uniformly prevalent between different groups of women. In terms of our study, this means that all woman should be empowered and that other social determinants such as wealth should be uniformly available to all women. However, in Ethiopia, empowerment of women is far from universal and has been linked in the literature with higher usage of maternal health services [16, 17]. The study's findings would provide information to policymakers that would help them comprehend how equity in maternal health care utilization is affected by improvements in women's empowerment and the fair distribution of other social elements.

To this end utilizing recently created validated indicators of women's empowerment, the study's objectives was to assess women's empowerment related disparities in the consumption of four maternal health care services. We also aimed to dissect the disparities to see which health care drivers were responsible for the observed disparities.

## Methods

### Setting of the study

Ethiopia, a landlocked country in Africa's horn, is the continent's second-most populous country. 115 million people live in Ethiopia. In this culturally diverse region, there are more than 80 ethnic groups of different origins, such as Cushitic and Afro-Asiatic [18].

With a growth rate of 6.3% in Fiscal Year (FY) 2020/21, Ethiopia's economy is expanding at the quickest rate in the region, with a gross national income of $890 per person. By 2025, Ethiopia hopes to become a lower-middle-income country. Ethiopia's economy has experienced some of the quickest growth worldwide during the past 15 years (at an average of 9.5 percent per year). Growth was fueled by capital accumulation, particularly through investments in public infrastructure, among other things. Because of the advent of COVID-19 pandemic, gross domestic product growth in Ethiopia decelerated in FY2019–20 and even more in FY2020–21, with industry and service growth slowing to single digits. But the epidemic did not seriously disrupt agriculture, where more than 70% of people work, nevertheless, and its share to the economy actually marginally increased in the FY2020/21 compared to the year before. Using the 2019 Home-Grown Economic Reform Agenda as a foundation, the government has unveiled a new 10-Year Development Plan that will cover the years 2020/2021–2029/2030. The goal of the strategy seeks to continue the impressive growth that was attained during

the Growth and Transformation Plans of the previous ten years while easing the transition to an economy controlled by the private sector. Additionally, it intends to boost competitiveness and productivity in vital industries that support growth (such as energy, logistics, and telecommunications), enhance the business culture, and correct macroeconomic inequities [19].

Ethiopia, like the rest of the globe, has been dealing with a broad spectrum of social, economic, and political issues that have a bearing on its advancement. The livelihoods of farmers and pastoralists as well as the security of food supply are threatened by recurrent extreme weather conditions and the protracted impacts of climate change. The worst drought in four decades will hit 7 million people hard in the south and east of the nation in 2022. There have been more conflicts, which has had a massive impact on people's lives, way of life, and facilities [19].

One of the lowest Human Development Indexes (HDI) in the world, Ethiopia's score is somewhat higher than the mean for low-income countries but still among the lowest in the world. It is also lower than the region of Sub-Saharan Africa. The HDI's value of 0.38 indicates that economic and social prosperity are not high. Ninety percent of children live in poverty in terms of education, while 37% of children under 5 have stunted growth. The largest locust infestation in Ethiopia in years erupted in 2020, and its ramifications have threatened agricultural production and living for multitudes of Ethiopians. Because of the strain that the nation's expanding population is putting on the labor market's capacity to accommodate workers, it is vital to enhance existing positions while also adding enough new ones [19].

Ethiopia's three-tiered health service delivery system offers primary, secondary, and tertiary level care. With Maternal and Child Health (MCH) service as the main focal area in addition to other areas of importance, the Health Extension Program (HEP), which is a part of the primary health care unit, seeks to bring healthcare closer to where people live. Between 2000 and 2017, maternal mortality in Ethiopia decreased by more than 50%. The drop in maternal deaths is brought on by both the availability of emergency obstetric care and the greater utilization of maternal care services [20, 21].

## Data, sample, and sampling procedure

We made use of information from the four waves of the Ethiopia Demographic and Health Surveys (EDHS), which were conducted in 2000, 2005, 2011, and 2016. Since its establishment in in 1984, the Demographic and Health Survey (DHS) program has offered technical support to over 350 surveys in more than 90 countries. The DHS Program's major goal is to enhance the gathering, analysis, and communication of demographic, health, and nutrition statistics as well as to make it easier to utilize these data for program administration, planning, and decision. In low- and middle-income nations, where there are few other or no ways to obtain data, the program has shown to be highly valuable for data gathering and analysis. Data that can be compared between nations can be collected thanks to the DHS Program. Standard model questions have been established and are being utilized by participating nations to do this [22].

The EDHS is the nationally representative household-based cross-sectional survey that is a part of the worldwide DHS project. Among the many issues covered by the EDHS are maternal health, women's empowerment, anemia, child health, domestic violence, nutrition, family planning, HIV/AIDS, Malaria, maternal and child mortality. Men of reproductive age (15–59 years) and children under the age of five are also included in the program, though women in the reproductive age range (15 to 49 years) make up the majority of the EDHS questions. Because the DHS only makes available anonymised data, the authors of this study were unable to acquire information that may have been used to identify specific individuals.

Data are gathered using both common questionnaires and questionnaires tailored to the country's specific situation after verbal informed consent was taken from each woman. Interviewers then sign on the consent form on the behave of the respondents. For the 2000 EDHS and the 2005 EDHS, participants were recruited and data were collected from them between early February 2000 and mid-June 2000, and between April 27 and August 30, 2005, respectively. For the 2011 EDHS, it occurred between 27 December 2010 and 3 June 2011, while for the 2016 EDHS, it happened from 18 January 2016 and 27 June 2016.

The surveys are sufficiently similar to allow fair comparison between them, even if the EDHSs have been modified over time to adapt for changing demographic demands [22–26].

The EDHS sample is intended to generate accurate estimates for key indicators for the nation as a whole, for urban and rural areas, as well as for the nine regions and two city administrations, from which the data were gathered. In the pertinent sections of the respective final documents that contain the report from them, the size of samples and participant selection procedure in the EDHS have been detailed in depth [23–26]. Succinctly, the samples were picked using a two-stage stratified cluster sampling procedure. Every region in the nation was originally divided into urban and rural sections. Following that, samples were taken in two stages separately from each stratum. Clusters were chosen in the first stage systematically using the probability proportional to size method, where the clusters serve as the primary sample units or enumeration areas. The 2007 Ethiopia Population and Housing Census (PHC) provided the sampling frame for the first stage of the 2011 and 2016 EDHS rounds and contained information about the enumeration areas. However, the sampling frame for the EDHS in 2000 and 2005 came from the 1994 PHC.

A trade-off must be made between the required survey precision and the available budget when determining the ideal sample size. In DHS surveys, the number of households to be selected is constant for both urban and rural areas for simplicity, with a few exceptions, because the number of PSUs required to get the required number of individuals depends on the number of houses to be selected in each selected PSU. While the cluster size may not have much of an impact on the sampling error if the second-stage sample size is fixed. The ideal sample size explicitly depends on the cost ratio and the intracluster correlation (ICC) [27]. The computation of the ideal sample size ultimately boils down to calculating the ICC. The optimal sample take is an increasing function of cost ratio and a decreasing function of ICC. Researchers who are interested can refer to the material for a detailed explanation of Aliaga A. and Ren R.'s (2006) determination of the ideal sample size [27].

A predetermined number of households were systematically selected from each enumeration area for the second stage. All eligible women in the selected households are included in the study. In 2000, 2005, 2011, and 2016, consecutively, 15,367, 14,070, 16,515, and 15,683 women between the ages of 15 and 49 were surveyed; the corresponding response rates were 97.8, 95,6, 95.0, and 94.6%.

## Variables

**Outcome variables.**   Early skilled first antenatal care (ANC), at least four skilled ANC, quality ANC, and skilled postnatal care (PNC) within two days of birth are the study's outcome variables. If the services are delivered by a doctor, nurse, midwife, or health officer, they are referred to as skilled services. Women are referred to as having early ANC if they started their first ANC within the first three months of their pregnancy; otherwise, they are referred to as having no early ANC. According to the World Health Organization's targeted ANC model, mothers who have received ANC at least four times are considered as having received the required number of visits (WHO). A new ANC model, however, with a minimum of eight

ANC visits, was established by WHO in 2016 [1]. Since the data were generated prior to the publication of the new model, we followed the previous ANC model of at least four visits in this study.

By including the particular ANC services or materials that a pregnant woman should receive during her ANC session, an indicator of quality ANC was formed. The accepted standard of care is that a pregnant woman should receive all of the following services during an ANC appointment: a blood pressure check, blood and urine tests, information on potential pregnancy issues, nutrition advice, and assistance in creating a birth plan. However, the study was limited to the three services since only three of the services—blood pressure measurement, blood testing, and urine testing—were consistently documented in all of the surveys. As a result, getting all three of these services counted as quality ANC. Services and materials supplied during the ANC session may act as a stand-in for judging the quality of the ANC and this practice has been documented elsewhere [28, 29]. PNC is defined as getting services within two days of delivery. All of the outcome variables are binary, with 1 indicating that the mother received the services and 0 indicating that she did not.

**Equity stratifier.** Investigating disparities in the utilization of maternal health care services requires the use of an equity stratifier, which in this case is women's empowerment. We employed a recently constructed indicator called "SWPER Global," which functions as a helpful overall indicator of women's empowerment and was developed for use in LMICs [30]. The index measures three distinct areas: decision-making, social independence, and attitude toward violence. The 14 DHS-available elements on which the domains are based are as follows:

1. Beating not justified if wife goes out without telling husband

2. Beating not justified if wife neglects the children

3. Beating not justified if wife argues with husband

4. Beating not justified if wife refuses to have sex with husband

5. Beating not justified if wife burns the food

6. Frequency of reading newspaper or magazine

7. Woman education

8. Age of respondent at cohabitation

9. Age of respondent at first birth

10. Age difference: woman's minus husband's age

11. Education difference: woman minus husband's years of schooling

12. Who usually decides on respondent's health care

13. Who usually decides on large household purchases

14. Who usually decides on visits to family or relatives

The social independence domain primarily consists of factors that might help women achieve their objectives, such as educational achievement, information access, age at significant life events, and spouse asset disparities and information access. On the other side, the decision-making domain includes topics like the involvement of women in household decisions. The notion of intrinsic agency is intimately tied to the attitudes toward violence domain, serving as a stand-in for the woman's assimilation of gendered cultural mores acceptability of violence.

The authors of the SWPER Global applied the technique of Principal Component Analysis (PCA) to drive the scores in the formation of the indices. Then, using the codes supplied by the authors of the SWPER Global, the authors of this paper directly compute the three indices. The three indices are each divided into low, medium, and high levels of empowerment, which converts the domains into ordinal type data and makes them ideal for assessing inequality. The creation of the indices has been detailed in full elsewhere [30].

**Independent and confounding variables.** Maternal age at pregnancy and at birth, place of residence, region, mother's education, partner's education, mother's religion, media exposure, wealth index, mother's occupation, partner's occupation, survey timing and Ethnicity are the independent or confounding variables in the study. We relied heavily on the conceptual, rather than statistical, understandings to determine whether the variables could potentially be confounders by using the three criteria of confounding. We avoided reliance on p-values to detect confounding variables as the sheer existence of statistically significant findings does not serve this objective [31]. We still included variables that are not confounders, but must be independent predictors of the maternal health care services as doing so would improve precision of the estimations for the focal independent variable (equity stratifier), that is, women's empowerment.

Mothers are divided into two categories based on their age at pregnancy or at birth: under 20 years and 20 years and older. Age at delivery is used for PNC outcome whereas age at pregnancy is used for ANC care outcome variables. Mother's and her partner's education is broken down into no education, primary, secondary, and higher education; mother's religion is broken down into Orthodox, Protestant, Muslim/other; media exposure is broken down into no exposure, one media exposure, two media exposures, and three media exposures; wealth is broken down into the poorest, poorer, middle, richer, and richest; and mother's and partner's occupation status is broken down into yes (the mother or partner has a occupation) and no. The survey variable indicates when the study was conducted (2000, 2005, 2011, and 2016) [30]. The "year" during which the surveys were conducted was included in our model to capture the changes in use of the services over time. Ethnicity is grouped into 20 categories. Finally, the three domains of women's empowerment themselves were included in respective regression models to capture their contribution to the inequalities. The decision-making domain is not available for the 2000 EDHS and disparities related with this domain were restricted to the three surveys. The choices on the grouping of the variables are made based on a number of considerations such as experience or field expertise (for example, it is known that utilization of maternal health care services varies by whether or not the receipt is adolescent mother), works of prior studies on this same area, and sample size.

## Statistical analysis

We looked into the inequality on a large sample after pooling the data from the four rounds of the EDHS (via appending), which caused the data size to increase significantly. We used concentration index (CI) and concentration curve [32] to present inequalities in the distributions of the maternal health care services across the different groups of women's empowerments. While concertation curve does help identify whether inequalities exist in the service, the CI has a use in quantifying the degree of variation in a health indicator (maternal health care services) among individuals who are rated by a certain socioeconomic factor, such as women's empowerment.

The concertation curve and CI are related measures of inequalities in health variable, and CI is twice the area between the concentration curve and the line of equality (the 45-degree line). The CI is bounded between –1 and 1, inclusive. Typically, CI turns negative when the

health variable in question is disproportionately concentrated among the disadvantaged categories of the ranking variable (women's empowerment in this study) and positive when the health variable is primarily concentrated among the advantaged/better categories of the ranking variable, i.e., higher empowerment in this study. In terms of the concentration curve, this means that the curve lies respectively above or below the line of no inequality. When no inequality exits, the CI becomes zero. Since it combines the three characteristics of a good inequality measure—reflecting the socioeconomic dimension of a health inequality, reflecting the entire range of the population studied, and being sensitive to changes in the distribution of each subgroup of the ranking variable—it has been suggested that CI is one of the most appropriate measures of inequality in health. Mathematically, the concentration index is defined as

$$CI = \frac{2}{\mu} cov(h, r) \tag{1}$$

Where h denotes the health variable (maternal health care services), $\mu$ is mean (proportion) of the health variable, r denotes the ranking variable (women's empowerment) and *cov* denotes covariance between h and r.

In order to accommodate the various characteristics of health variables to be monitored, different variations of the CI have been devised. The standard concentration index is appropriate for health indicator variables measured on ratio-scale. However, most health variables are binary in nature and are not measured on ratio scale, rendered the standard concentration index non-suitable index for inequality measure. For binary outcome variables like maternal health care services utilization, where values of the standard CI do not fall between -1 and 1, the Erreygers normalized concentration index has been suggested in the literature [33]. Therefore, we made use of the Erreygers normalized concentration index (ECI) in this investigation. It measures absolute inequality and meets the mirror requirement, meaning that an inequality in attainment is equivalent to an inequality in shortfalls, with the exception that their signs are different. This trait is significant since it eliminates the need to calculate CI separately for attainments and shortfalls. Erreygers normalized concentration index is formally defined as

$$ECI = \frac{4\hat{y}}{y^{max} - y^{min}} CI \tag{2}$$

Where $y^{max} - y^{min}$ is the range of the health variable, which is 'one' for of binary variables, $\hat{y}$ denotes the mean of the health variable (maternal health care services). The *conindex and clorenz* Stata commands were used to calculate the index and to do the concentration curve [34].

After computing the ECI and curve, a decomposition analysis was conducted to determine the impact of different predictor factors to the inequality in the use of maternal health care services based on women's empowerment. The regression-based decomposition analysis technique proposed by O'Donnell O et al was used for this purpose [32]. Particularly, we fitted generalized linear model with the binomial family and logit link because the response variables are binary. Elasticity, concentration indexes, and contributions from each explanatory factor included in the regression model are all produced via the decomposition procedure. Elasticity is the percentage change in the dependent variables (maternal health care services) that results from a change of one unit in the independent variables. An increase or decrease in the prevalence of maternal health care services in relation to a change in the independent variable is indicated by a positive or negative sign in the elasticity, respectively [35].

The concentration index is a representation of the exposure variables' concentration index in relation to the three areas of women's empowerment. Depending on whether the

value is positive or negative, the prevalence of the exposure variables is more common among the higher or lower categories of women's empowerment, respectively. Lastly, the percentage contribution shows how much each variable in the model has contributed overall to the disparity in maternal health care services. The observed inequality rises or falls, depending on whether a variable's percentage contribution is positive or negative, respectively [32].

Due to the intricate sampling procedure utilized to gather the EDHS data, the three survey design components—weight, stratification, and clustering—were taken into account during analysis using the Stata module "svy." The analysis was done based on the pooled data drawn from the four waves of the EDHS. The weight variable was rescaled in the pooled dataset so that each survey counts equally in the analysis. In a pooled analysis of more than one surveys from a country, it could be possible that analysts would run the risk of having same names for different strata and clusters. To prevent such a problem, we defined distinct strata and cluster variables in the pooled dataset. Stata code has been supplied as a S1 File for the preparation of the design elements in a pooled DHS data and their use in the analysis to improve the reproducibility of our research. Missing and do not know responses were handled based on the 2018 guide to DHS statistics [22]. We used the missing and do not know responses in the denominator when we create variables that have such responses. Further, we recoded them into the 'less advantageous' category of that variable (for example, into "no occupation" category for the occupation variables). We used the STROBE reporting guideline along with an article that explains how the checklist was used to convey our findings [36]. All of the analyses were done in Stata 16 (Release 16. College Station, TX: StataCorp LLC).

## Ethics statement

For the protection of respondents' privacy in all EDHS surveys, the DHS program upholds tight criteria. ICF Institutional Review Board (IRB) has evaluated and given the go-ahead for the procedures and questionnaires. The ICF IRB and the IRB in Ethiopia both examined the EDHS protocols that are particular to Ethiopia. The respondents were asked to verbally consent after being informed about the purpose of the study.

## Results

The characteristics of the sampled women are given in Table 1. In the pooled data, women aged 20 or older make up more than 85% of the sample population. About 1.2% of the respondents are in the age group of eight years and 3 months and 15 years. Almost eight in ten women resided in rural areas. A little more than 1% of women reported having three different types of media exposure, including television, radio, and newspaper reading.

## Inequalities in the use of maternal health services by attitude towards violence

The ECIs for the maternal health services are 0.240 (95% CI 0.207, 0.273) for quality ANC; 0.083(95%CI 0.071, 0.095) for early ANC visit; (0.122; 95%CI 0.106, 0.138) for the at least four ANC visits and (0.067; 95%CI 0.053, 0.081) for PNC within 2 days. The ECI for all the four maternal health care services are positive, showing that the services were inequitably utilized by women to the advantage of highly empowered women. Similarly, the concentration curves, as presented in Fig 1, lie below the line of no difference, suggesting that the services were mainly concentrated among highly empowered women.

**Table 1. Sociodemographic characteristics of the sample, 2000–2016 EDHS.**

| Variables | %(95%CI) |
| --- | --- |
| **Maternal age at pregnancy** | |
| > = 20 | 85.08 (84.52, 85.63) |
| < 20 | 14.92 (14.37,15.48) |
| **Place of residence** | |
| Urban | 20.5 (19.57, 21.46) |
| Rural | 79.5 (78.54,80.43) |
| **Region** | |
| Tigray | 6.68 (6.37, 7.01) |
| Afar | 0.97 (0.89, 1.06) |
| Amhara | 25.03 (24.21,25.88) |
| Oromia | 36.75 (35.8, 37.7) |
| Somali | 2.38 (2.14, 2.65) |
| Benishangul | 1.0 (0.91, 1.10) |
| SNNPR | 20.81(20.17, 21.46) |
| Gambella | 0.32 (0.27,0.37) |
| Harar | 0.27 (0.26, 0.28) |
| Addis Ababa | 5.3 (5.05, 5.56) |
| Dire Dawa | 0.50 (0.47, 0.53) |
| **Religion** | |
| Orthodox | 47.62 (45.73, 49.52) |
| Protestant | 20.03 (18.52, 21.64) |
| Muslim | 29.12 (27.01, 31.31) |
| Traditional/other | 3.23 (2.7,3.86) |
| **Media exposure** | |
| No exposure | 77.02 (76.11, 77.9) |
| Exposed to one media | 16.43 (15.77, 17.1) |
| Exposed to 2 media | 5.46 (5.09, 5.86) |
| Exposed to 3 media | 1.1 (0.96, 1.25) |
| **Wealth** | |
| Poorest | 17.68 (16.66, 18.74) |
| Poorer | 18.47(17.72,19.25) |
| Middle | 18.96 (18.22,19.72) |
| Richer | 19.38 (18.42, 20.38) |
| Richest | 25.51 (24.43, 26.62) |
| **Women occupation** | |
| No occupation | 49.0 (47.73, 50.27) |
| Has occupation | 51.0 (49.73, 52.27) |
| **Partner occupation** | |
| No occupation | 3.07 (2.74, 3.43) |
| Has occupation | 96.93 (96.57, 97.26) |
| **Maternal education** | |
| No education | 59.92 (58.86, 60.98) |
| Primary | 27.75 (26.91, 28.6) |
| Secondary | 9.34 (8.86, 9.85) |
| Higher | 2.99 (2.66, 3.35) |
| **Partner education** | |
| No education | 56.78 (55.69, 57.86) |

(*Continued*)

**Table 1.** (Continued)

| Variables | %(95%CI) |
|---|---|
| Primary | 30.25 (29.35, 31.18) |
| Secondary | 9.13 (8.63, 9.66) |
| Higher | 3.83 (3.48, 4.23) |
| **Women's empowerment** | |
| **Attitude towards violence** | |
| Low | 58.79 (57.59, 59.97) |
| Medium | 19.78 (19.04,20.56) |
| High | 21.43(20.54,22.34) |
| **Social independence** | |
| Low | 65.28(64.34, 66.21) |
| Medium | 25.45(24.71, 26.19) |
| High | 9.271 (8.783, 9.784) |
| **Decision making** | |
| Low | 11.54(10.74, 12.4) |
| Medium | 31.83 (30.8, 32.89) |
| High | 56.63 (55.26, 57.98) |
| **Ethnicity** | |
| Tigrie | 2.31 (2.0, 2.66) |
| Amhara | 7.99 (7.44, 8.57) |
| Oromo | 8.98 (8.15, 9.90) |
| Afar | 16.05 (15.15, 17) |
| Somali | 8.87 (8.06, 9.76) |
| Guragie | 0.73 (0.52, 1.02) |
| Sidama | 1.05 (0.63, 1.72) |
| Wolaita | 0.82 (0.48, 1.39) |
| Berta | 0.23 (0.12, 0.45) |
| Anywak | 0.02 (0.01, 0.03) |
| Donga | 2.51 (1.93, 3.27) |
| Ari | 0.88 (0.6, 1.27) |
| Mossiye | 16.74 (15.82, 17.71) |
| Benchi | 0.12 (0.05, 0.40) |
| Bodi | 8.12 (7.41, 8.90) |
| Qewama | 3.40 (3.16, 3.67) |
| Burji | 0.52 (0.31, 0.86) |
| Bena | 0.58 (0.44, 0.75) |
| Chara | 1.72 (1.59, 1.87) |
| others | 18.35 (17.14, 19.61) |

## Inequalities in the use of maternal health services by social independence

The ECIs are 0.20 (95% CI 0.169, 0.231) for quality ANC; 0.067 (95%CI 0.055, 0.079) for early ANC visit; 0.122 (95%CI 0.108, 0.136) for at least four ANC visits and 0.071(95%CI 0.057, 0.085) for PNC within 2 days. The ECIs are all positive and indicate that more of the services were utilized by women from higher empowerment status. The results of the concentration curves (Fig 2) also demonstrate that the utilization of the services was favored by highly empowered women.

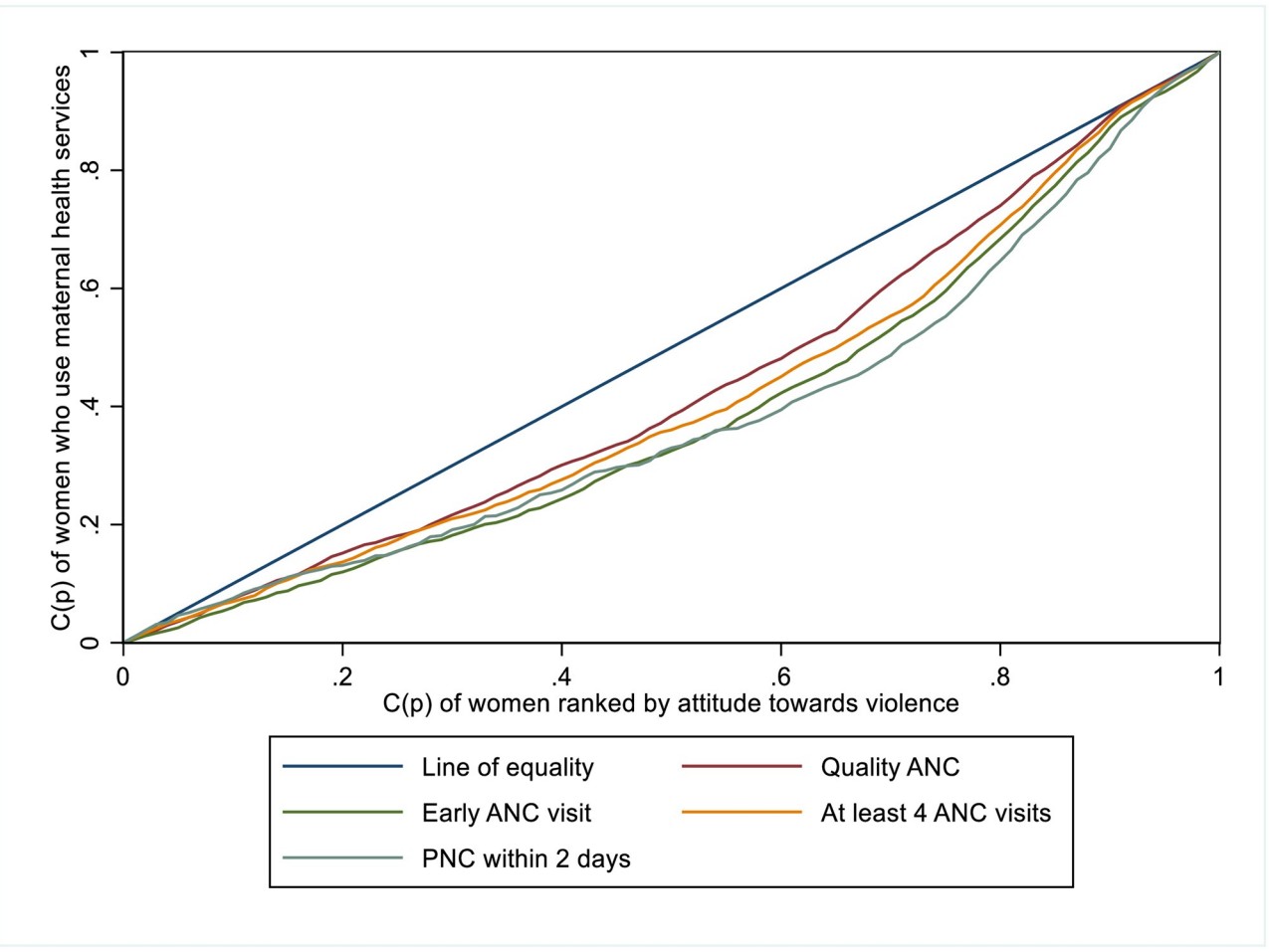

**Fig 1. Concentration curves of maternal health services based on attitude towards violence domian of women's empowerment.** Note: C(p) refers to cumulative proportion.

### Inequalities in the use of maternal health services by decision making

The ECIs are 0.122 (95% CI 0.087, 0.157) for quality ANC; 0.065(95%CI 0.053, 0.077) for early ANC visit; 0.092(95%CI 0.074, 0.11) for at least four ANC visits and 0.057 (95%CI 0.043, 0.071) for PNC within 2 days. Values of the ECIs indicate the unequal distribution of the services to the disadvantage of the poorly empowered women. However, the concentration curves (Fig 3) do fluctuate between below and above the line of no difference.

The decomposition of inequalities in the utilization of maternal health care services based on women's empowerment is shown in Tables 2–5. The tables contain information on elasticities, concentration indices and percent contributions (%) of each of the explanatory variables used in the regression analysis.

The elasticity findings for the different exposure variables are different, indicating that the extent of influence on the utilization of maternal health care services of a unit change in them is different. For example, rural residence has an elasticity of -0.207 for the attitude towards violence-based inequality in the use of quality ANC, which translates to mean that a change from urban to rural residence led to a 0.21 percent decrease in the use of the service. The elasticity of the richest wealth for the PNC service ranged from 0.24 for the attitude towards violence and social independence domains to 0.28 for decision making domain, indicating that for a change

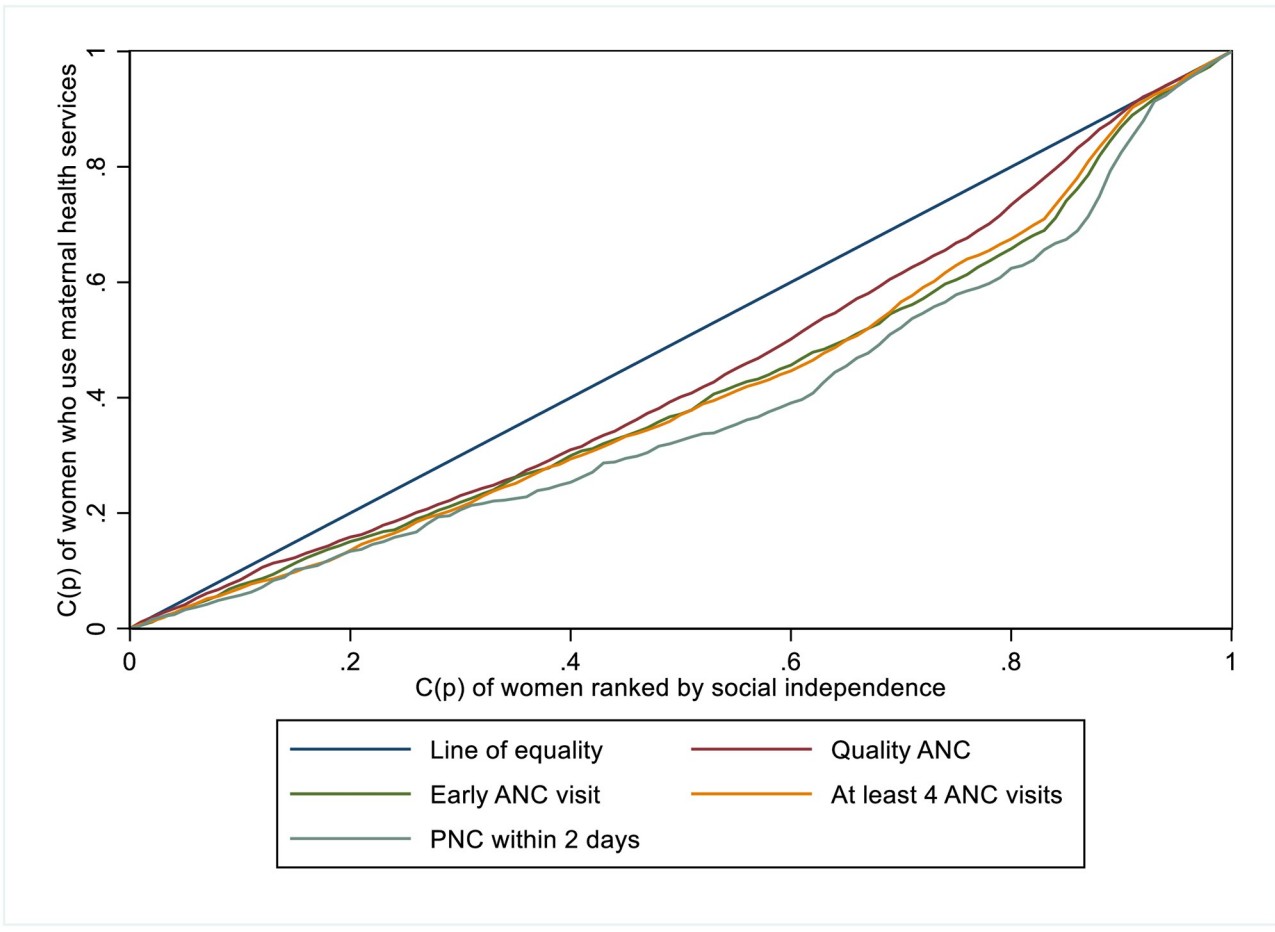

**Fig 2. Concentration curves of maternal health services based on social independence domian of women's empowerment.** Note: C(p) refers to cumulative proportion.

of women's status from poorest group to richest group, the use of the service increases by 0.24 to 0.28%.

The women's empowerment-based inequalities in the uptake of the maternal health care services were mainly driven by rural residence, regions including Addis Ababa city, SNNPR and Amhara, wealth status of the households, educational status of the mother and her partner, the time in which the surveys were conducted, the three domains of women's empowerment themselves, media exposure and maternal religion. These same variables were responsible for the largest part of the inequality though their contribution differed slightly depending on the type of maternal health care services considered and the type of women's empowerment domains used to calculate the inequalities with.

Most of the variables contributed to help the observed disparities widen in the use of the services across women positioned in different categories of women's empowerment. Rural residence, for example, contributed 6.9, 7.6 and 8.5%, respectively, to the inequality in the usage of quality ANC, measured based on women's attitude to violence, social independence and decision-making domains of women's empowerment. Richest wealth, on the other hand, has a contribution ranged between 8% and 13%, for the attitude to violence and social independence driven inequality in the usage of quality ANC.

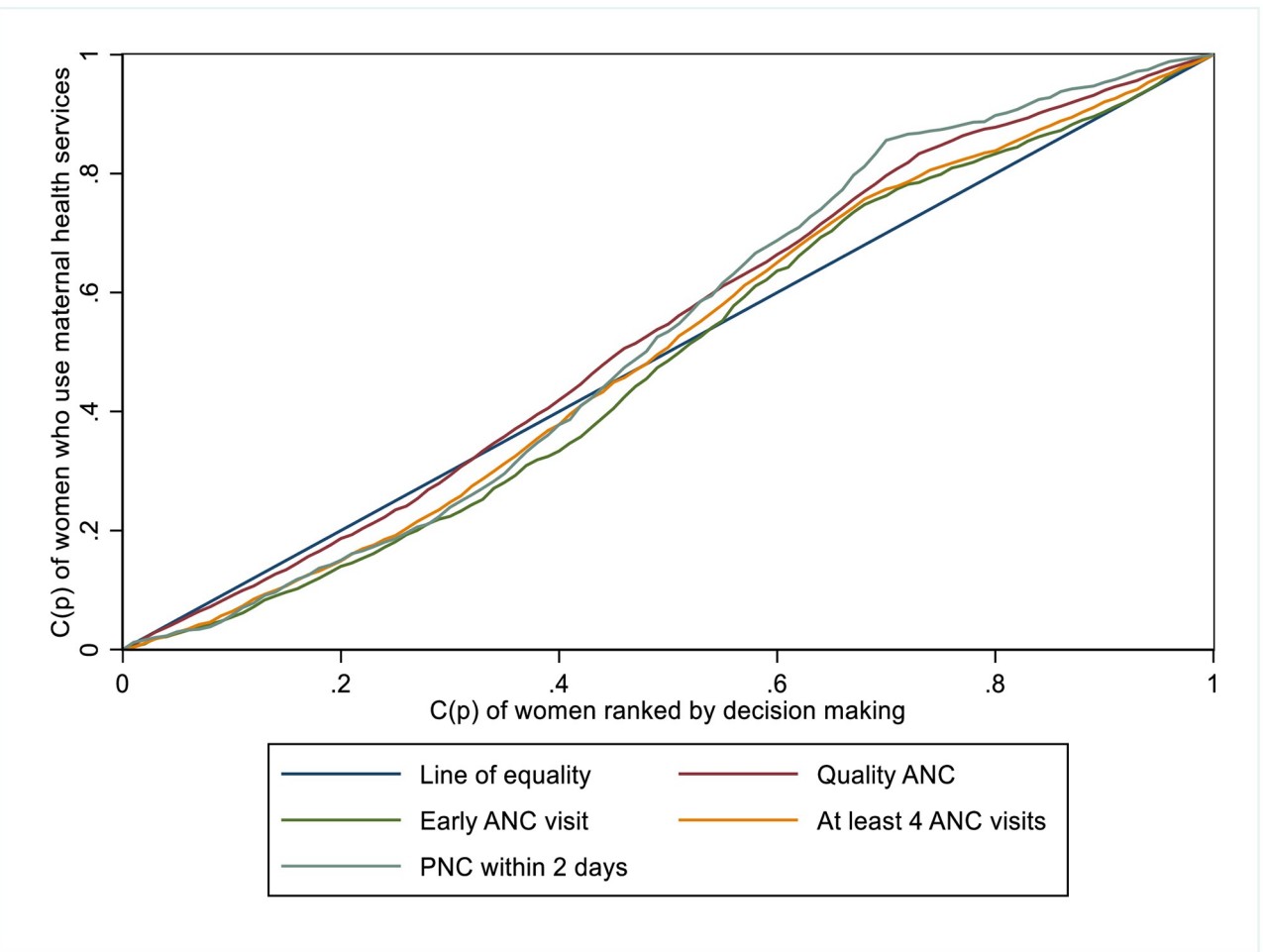

**Fig 3. Concentration curves of maternal health services based on decision making domian of women's empowerment.** Note: C(p) refers to cumulative proportion.

Some variables contributed to decrease the inequality in the utilization of the services. For example, wealth index other than richest category, has negative contribution, indicating that it contributed to narrowing the inequalities in the utilization of all the maternal health care services. Still some other variables have both negative and positive contributions to the same maternal health care services depending on the categories of the variables.

## Discussion

Utilizing the three domains of women's empowerment as an equity stratifier, our study intended to quantify disparities in the utilization of maternal health care services. Further, the study attempted to dissect the observed inequalities in the use of the services between highly and poorly empowered women into underlying layers of inequalities. The study's foundation was pooled data taken from the four rounds of the EDHS, with a particular emphasis on women in the reproductive age range. The study found that higher levels of women's empowerment were linked to greater concentrations of maternal health care services, meaning that use of the maternal health care services was unevenly distributed across the various groups of women's empowerment domains to the benefit of highly empowered women.

**Table 2. Decomposition of the inequalities in the usage of quality ANC by women's empowerment, 2000–2016 EDHS.**

| Variables | Domains of women's empowerment | | | | | | | | |
|---|---|---|---|---|---|---|---|---|---|
| | Attitude to violence (N = 10,129) | | | Social independence (N = 10,129) | | | Decision making (N = 8,114) | | |
| | Elasticity | CI | % | Elasticity | CI | % | Elasticity | CI | % |
| **Maternal age at pregnancy** | | | | | | | | | |
| > = 20 (ref) | | | | | | | | | |
| < 20 | -0.007 | -0.008 | 0.034 | -0.007 | -0.152 | 0.773 | -0.01 | -0.028 | 0.337 |
| **Place of residence** | | | | | | | | | |
| Urban(ref) | | | | | | | | | |
| Rural | -0.21 | -0.052 | 6.943 | -0.212 | -0.047 | 7.65 | -0.214 | -0.032 | 8.44 |
| **Region** | | | | | | | | | |
| Tigray (ref) | | | | | | | | | |
| Afar | -0.003 | -0.052 | 0.086 | -0.003 | -0.034 | 0.068 | -0.003 | -0.067 | 0.239 |
| Amhara | -0.082 | -0.043 | 2.267 | -0.084 | -0.143 | 9.148 | -0.085 | 0.087 | -9.203 |
| Oromia | -0.049 | -0.009 | 0.27 | -0.052 | 0.012 | -0.471 | -0.047 | -0.005 | 0.313 |
| Somali | 0 | 0.027 | 0.005 | 0 | 0.048 | 0.004 | -0.001 | -0.202 | 0.154 |
| Benishangul | -0.003 | 0.053 | -0.094 | -0.003 | -0.049 | 0.106 | -0.003 | -0.06 | 0.225 |
| SNNPR | -0.076 | -0.035 | 1.676 | -0.078 | 0.084 | -5.002 | -0.084 | -0.117 | 12.239 |
| Gambella | 0 | 0.106 | -0.028 | 0 | 0.032 | -0.011 | -0.001 | -0.11 | 0.123 |
| Harar | 0.001 | 0.246 | 0.156 | 0.001 | 0.169 | 0.127 | 0.001 | 0.142 | 0.171 |
| Addis Ababa | 0.041 | 0.547 | 14.452 | 0.041 | 0.47 | 14.801 | 0.044 | 0.251 | 13.884 |
| Dire Dawa | 0.002 | 0.255 | 0.253 | 0.002 | 0.231 | 0.268 | 0.002 | 0.102 | 0.216 |
| **Religion** | | | | | | | | | |
| Orthodox (ref) | | | | | | | | | |
| Protestant | -0.036 | -0.024 | 0.551 | -0.037 | 0.086 | -2.408 | -0.042 | -0.056 | 2.959 |
| Muslim | 0.012 | -0.027 | -0.198 | 0.012 | -0.022 | -0.199 | 0.015 | -0.054 | -1.044 |
| Traditional/other | 0 | -0.069 | 0.005 | 0 | 0.003 | 0 | 0 | -0.151 | 0.042 |
| **Media exposure** | | | | | | | | | |
| No exposure (ref) | | | | | | | | | |
| Exposed to 1 media | 0.011 | 0.141 | 1.019 | 0.011 | 0.115 | 0.99 | 0.009 | 0.073 | 0.781 |
| Exposed to 2 media | 0.01 | 0.422 | 2.761 | 0.01 | 0.456 | 3.501 | 0.011 | 0.207 | 2.844 |
| Exposed to 3 media | 0.002 | 0.597 | 0.635 | 0.002 | 0.717 | 0.937 | 0.002 | 0.293 | 0.674 |
| **Wealth** | | | | | | | | | |
| Poorest (ref) | | | | | | | | | |
| Poorer | 0.006 | -0.08 | -0.281 | 0.006 | -0.066 | -0.292 | 0.011 | -0.045 | -0.596 |
| Middle | 0.019 | -0.057 | -0.675 | 0.019 | -0.054 | -0.791 | 0.021 | -0.036 | -0.932 |
| Richer | 0.019 | -0.025 | -0.292 | 0.019 | -0.024 | -0.343 | 0.027 | 0.023 | 0.773 |
| Richest | 0.051 | 0.239 | 7.699 | 0.051 | 0.218 | 8.521 | 0.066 | 0.147 | 12.092 |
| **Women occupation** | | | | | | | | | |
| No occupation (ref) | | | | | | | | | |
| Has occupation | -0.017 | 0.015 | -0.158 | -0.018 | 0.008 | -0.108 | -0.013 | 0.073 | -1.178 |
| **Partner occupation** | | | | | | | | | |
| No occupation (ref) | | | | | | | | | |
| Has occupation | -0.1 | -0.003 | 0.17 | -0.1 | -0.002 | 0.145 | -0.077 | -0.006 | 0.556 |
| **Year of survey** | | | | | | | | | |
| 2000 (ref) | | | | | | | | | |
| 2005 | 0.023 | -0.102 | -1.469 | 0.024 | -0.035 | -0.637 | ref | | |
| 2011 | 0.137 | 0.066 | 5.712 | 0.138 | 0.027 | 2.876 | 0.138 | -0.021 | -3.712 |
| 2016 | 0.255 | 0.142 | 23.185 | 0.258 | 0.066 | 12.963 | 0.275 | 0.124 | 42.752 |

(*Continued*)

**Table 2.** (Continued)

| Variables | Domains of women's empowerment | | | | | | | | |
|---|---|---|---|---|---|---|---|---|---|
| | Attitude to violence (N = 10,129) | | | Social independence (N = 10,129) | | | Decision making (N = 8,114) | | |
| | Elasticity | CI | % | Elasticity | CI | % | Elasticity | CI | % |
| **Maternal education** | | | | | | | | | |
| No education (ref) | | | | | | | | | |
| Primary | 0.033 | 0.078 | 1.663 | 0.03 | 0.14 | 3.221 | 0.034 | 0.032 | 1.358 |
| Secondary | 0.025 | 0.379 | 6.133 | 0.025 | 0.618 | 11.618 | 0.021 | 0.228 | 6.065 |
| Higher | 0.014 | 0.64 | 5.682 | 0.015 | 0.836 | 9.272 | 0.013 | 0.322 | 5.231 |
| **Partner education** | | | | | | | | | |
| No education(ref) | | | | | | | | | |
| Primary | 0.03 | 0.013 | 0.252 | 0.03 | 0.002 | 0.046 | 0.035 | -0.002 | -0.084 |
| Secondary | 0.017 | 0.186 | 2.068 | 0.018 | 0.203 | 2.744 | 0.024 | 0.095 | 2.904 |
| Higher | 0.006 | 0.488 | 1.924 | 0.006 | 0.524 | 2.544 | 0.009 | 0.235 | 2.631 |
| **Ethnicity** | | | | | | | | | |
| Tigrie (ref) | | | | | | | | | |
| Amhara | 0.003 | 0.165 | 0.332 | 0.004 | 0.032 | 0.1 | 0.001 | 0.189 | 0.316 |
| Oromo | -0.026 | 0.081 | -1.362 | -0.025 | 0.039 | -0.757 | -0.031 | 0.114 | -4.389 |
| Afar | -0.025 | -0.113 | 1.827 | -0.023 | -0.138 | 2.421 | -0.028 | 0.037 | -1.291 |
| Somali | -0.011 | 0.129 | -0.925 | -0.01 | -0.056 | 0.428 | -0.011 | 0.043 | -0.619 |
| Guragie | 0.001 | 0.35 | 0.191 | 0.001 | 0.162 | 0.123 | 0.001 | 0.073 | 0.096 |
| Sidama | 0 | 0.122 | -0.016 | 0 | -0.018 | 0 | 0 | -0.035 | 0 |
| Wolaita | -0.003 | 0.126 | -0.214 | -0.003 | 0.166 | -0.324 | -0.003 | 0.093 | -0.334 |
| Berta | -0.001 | 0.037 | -0.014 | -0.001 | -0.036 | 0.016 | -0.001 | -0.141 | 0.103 |
| Anywak | 0 | 0.228 | -0.008 | 0 | 0.111 | -0.005 | 0 | 0.075 | -0.003 |
| Donga | 0.002 | -0.02 | -0.026 | 0.002 | 0.116 | 0.216 | 0.005 | -0.143 | -0.84 |
| Ari | 0.000 | 0.242 | 0.021 | 0.000 | 0.254 | 0.041 | 0 | 0.057 | 0.023 |
| Mossiye | -0.037 | -0.112 | 2.657 | -0.035 | -0.025 | 0.662 | -0.02 | -0.128 | 3.197 |
| Benchi | 0.000 | 0.063 | 0.014 | 0.000 | -0.008 | -0.002 | 0.000 | 0.031 | 0.018 |
| Bodi | -0.017 | 0.081 | -0.859 | -0.016 | 0.05 | -0.598 | -0.019 | 0.000 | -0.001 |
| Qewama | -0.009 | -0.009 | 0.05 | -0.008 | -0.079 | 0.512 | -0.008 | 0.017 | -0.176 |
| Burji | 0.001 | -0.007 | -0.002 | 0.001 | 0.063 | 0.028 | 0.001 | 0.014 | 0.013 |
| Bena | -0.001 | -0.138 | 0.114 | -0.001 | 0.073 | -0.07 | -0.001 | -0.258 | 0.367 |
| Chara | -0.003 | 0.156 | -0.321 | -0.003 | 0.053 | -0.128 | -0.003 | 0.077 | -0.305 |
| others | -0.021 | -0.057 | 0.741 | -0.018 | 0.072 | -1.014 | -0.019 | -0.156 | 3.696 |
| **Women's empowerment** | | | | | | | | | |
| **Attitude towards violence** | | | | | | | | | |
| Low (ref) | | | | | | | | | |
| Medium | -0.006 | 0.374 | -1.435 | | | | | | |
| High | 0.007 | 0.786 | 3.636 | | | | | | |
| **Social independence** | | | | | | | | | |
| Low (ref) | | | | | | | | | |
| Medium | | | | 0.012 | 0.56 | 5.127 | | | |
| High | | | | 0.000 | 0.907 | -0.286 | | | |
| **Decision making** | | | | | | | | | |
| Low (ref) | | | | | | | | | |
| Medium | | | | | | | 0.008 | -0.451 | -4.282 |
| High | | | | | | | 0.008 | 0.434 | 4.077 |

**Table 3. Decomposition of the inequalities in the usage of early ANC by women's empowerment, 2000–2016 EDHS.**

| Variables | Domains of women's empowerment | | | | | | | | |
|---|---|---|---|---|---|---|---|---|---|
| | Attitude to violence (N = 25,911) | | | Social independence (N = 25,911) | | | Decision making (N = 19,480) | | |
| | Elasticity | CI | % | Elasticity | CI | % | Elasticity | CI | % |
| **Maternal age at pregnancy** | | | | | | | | | |
| > = 20 (ref) | | | | | | | | | |
| < 20 | -0.002 | -0.008 | 0.006 | 0.001 | -0.152 | -0.08 | 0.007 | -0.028 | -0.106 |
| **Place of residence** | | | | | | | | | |
| Urban(ref) | | | | | | | | | |
| Rural | -0.455 | -0.052 | 10.784 | -0.457 | -0.047 | 12.165 | -0.519 | -0.032 | 9.472 |
| **Region** | | | | | | | | | |
| Tigray (ref) | | | | | | | | | |
| Afar | -0.005 | -0.052 | 0.111 | -0.005 | -0.034 | 0.087 | -0.006 | -0.067 | 0.24 |
| Amhara | -0.124 | -0.043 | 2.459 | -0.122 | -0.143 | 9.839 | -0.173 | 0.087 | -8.709 |
| Oromia | -0.179 | -0.009 | 0.706 | -0.175 | 0.012 | -1.18 | -0.277 | -0.005 | 0.86 |
| Somali | -0.007 | 0.027 | -0.083 | -0.006 | 0.048 | -0.159 | -0.009 | -0.202 | 1.047 |
| Benishangul | -0.008 | 0.053 | -0.197 | -0.008 | -0.049 | 0.218 | -0.011 | -0.06 | 0.379 |
| SNNPR | -0.078 | -0.035 | 1.236 | -0.075 | 0.084 | -3.532 | -0.105 | -0.117 | 7.104 |
| Gambella | 0.001 | 0.106 | 0.04 | 0.001 | 0.032 | 0.017 | 0.001 | -0.11 | -0.032 |
| Harar | 0.000 | 0.246 | 0.04 | 0.000 | 0.169 | 0.041 | 0.000 | 0.142 | 0.002 |
| Addis Ababa | -0.001 | 0.547 | -0.246 | 0.000 | 0.47 | -0.067 | -0.007 | 0.251 | -1.086 |
| Dire Dawa | 0.002 | 0.255 | 0.289 | 0.003 | 0.231 | 0.34 | 0.003 | 0.102 | 0.148 |
| **Religion** | | | | | | | | | |
| Orthodox (ref) | | | | | | | | | |
| Protestant | -0.102 | -0.024 | 1.107 | -0.105 | 0.086 | -5.111 | -0.119 | -0.056 | 3.87 |
| Muslim | -0.046 | -0.027 | 0.562 | -0.048 | -0.022 | 0.604 | -0.04 | -0.054 | 1.262 |
| Traditional/other | -0.017 | -0.069 | 0.531 | -0.017 | 0.003 | -0.031 | -0.017 | -0.151 | 1.446 |
| **Media exposure** | | | | | | | | | |
| No exposure (ref) | | | | | | | | | |
| Exposed to 1 media | 0.005 | 0.141 | 0.351 | 0.006 | 0.115 | 0.4 | 0.006 | 0.073 | 0.254 |
| Exposed to 2 media | 0.011 | 0.422 | 2.177 | 0.011 | 0.456 | 2.917 | 0.013 | 0.207 | 1.556 |
| Exposed to 3 media | 0.004 | 0.597 | 1.083 | 0.004 | 0.717 | 1.548 | 0.005 | 0.293 | 0.835 |
| **Wealth** | | | | | | | | | |
| Poorest (ref) | | | | | | | | | |
| Poorer | 0.04 | -0.08 | -1.474 | 0.04 | -0.066 | -1.487 | 0.053 | -0.045 | -1.366 |
| Middle | 0.064 | -0.057 | -1.657 | 0.064 | -0.054 | -1.944 | 0.08 | -0.036 | -1.668 |
| Richer | 0.085 | -0.025 | -0.959 | 0.086 | -0.024 | -1.149 | 0.1 | 0.023 | 1.331 |
| Richest | 0.169 | 0.239 | 18.368 | 0.171 | 0.218 | 21.008 | 0.19 | 0.147 | 16.183 |
| **Women occupation** | | | | | | | | | |
| No occupation (ref) | | | | | | | | | |
| Has occupation | 0.067 | 0.015 | 0.45 | 0.065 | 0.008 | 0.294 | 0.084 | 0.073 | 3.534 |
| **Partner occupation** | | | | | | | | | |
| No occupation (ref) | | | | | | | | | |
| Has occupation | -0.306 | -0.003 | 0.373 | -0.287 | -0.002 | 0.309 | -0.35 | -0.006 | 1.168 |
| **Year of survey** | | | | | | | | | |
| 2000 (ref) | | | | | | | | | |
| 2005 | 0.013 | -0.102 | -0.589 | 0.012 | -0.035 | -0.234 | ref | | |
| 2011 | 0.106 | 0.066 | 3.177 | 0.109 | 0.027 | 1.675 | 0.114 | -0.021 | -1.419 |
| 2016 | 0.191 | 0.142 | 12.44 | 0.196 | 0.066 | 7.277 | 0.198 | 0.124 | 14.221 |

(*Continued*)

**Table 3.** (Continued)

| Variables | Domains of women's empowerment | | | | | | | | |
|---|---|---|---|---|---|---|---|---|---|
| | Attitude to violence (N = 25,911) | | | Social independence (N = 25,911) | | | Decision making (N = 19,480) | | |
| | Elasticity | CI | % | Elasticity | CI | % | Elasticity | CI | % |
| **Maternal education** | | | | | | | | | |
| No education (ref) | | | | | | | | | |
| Primary | 0.095 | 0.078 | 3.395 | 0.098 | 0.14 | 7.719 | 0.106 | 0.032 | 1.932 |
| Secondary | 0.045 | 0.379 | 7.744 | 0.042 | 0.618 | 14.498 | 0.047 | 0.228 | 6.182 |
| Higher | 0.012 | 0.64 | 3.441 | 0.009 | 0.836 | 4.269 | 0.016 | 0.322 | 2.896 |
| **Partner education** | | | | | | | | | |
| No education(ref) | | | | | | | | | |
| Primary | 0.039 | 0.013 | 0.237 | 0.04 | 0.002 | 0.045 | 0.04 | -0.002 | -0.045 |
| Secondary | 0.03 | 0.186 | 2.557 | 0.031 | 0.203 | 3.515 | 0.037 | 0.095 | 2.021 |
| Higher | 0.014 | 0.488 | 3.056 | 0.014 | 0.524 | 4.232 | 0.012 | 0.235 | 1.653 |
| **Ethnicity** | | | | | | | | | |
| Tigrie (ref) | | | | | | | | | |
| Amhara | 0.036 | 0.165 | 2.695 | 0.036 | 0.032 | 0.645 | 0.048 | 0.189 | 5.241 |
| Oromo | -0.003 | 0.081 | -0.104 | -0.004 | 0.039 | -0.085 | 0.012 | 0.114 | 0.807 |
| Afar | 0.018 | -0.113 | -0.938 | 0.016 | -0.138 | -1.215 | 0.055 | 0.037 | 1.171 |
| Somali | -0.006 | 0.129 | -0.334 | -0.006 | -0.056 | 0.186 | 0.001 | 0.043 | 0.017 |
| Guragie | 0.000 | 0.35 | -0.063 | 0.000 | 0.162 | -0.044 | 0.000 | 0.073 | 0.017 |
| Sidama | 0.002 | 0.122 | 0.129 | 0.002 | -0.018 | -0.023 | 0.004 | -0.035 | -0.079 |
| Wolaita | -0.003 | 0.126 | -0.152 | -0.003 | 0.166 | -0.259 | -0.002 | 0.093 | -0.106 |
| Berta | 0.002 | 0.037 | 0.037 | 0.002 | -0.036 | -0.044 | 0.003 | -0.141 | -0.208 |
| Anywak | 0.000 | 0.228 | -0.003 | 0.000 | 0.111 | -0.002 | 0.000 | 0.075 | 0.000 |
| Donga | 0.007 | -0.02 | -0.062 | 0.007 | 0.116 | 0.442 | 0.01 | -0.143 | -0.809 |
| Ari | -0.001 | 0.242 | -0.123 | -0.001 | 0.254 | -0.173 | -0.001 | 0.057 | -0.026 |
| Mossiye | 0.031 | -0.112 | -1.58 | 0.029 | -0.025 | -0.406 | 0.042 | -0.128 | -3.123 |
| Benchi | 0.000 | 0.063 | 0.006 | 0.000 | -0.008 | -0.001 | 0.000 | 0.031 | 0.006 |
| Bodi | -0.011 | 0.081 | -0.411 | -0.011 | 0.05 | -0.319 | -0.002 | 0.000 | 0.000 |
| Qewama | -0.013 | -0.009 | 0.053 | -0.013 | -0.079 | 0.563 | -0.015 | 0.017 | -0.147 |
| Burji | -0.001 | -0.007 | 0.004 | -0.001 | 0.063 | -0.036 | -0.001 | 0.014 | -0.004 |
| Bena | -0.002 | -0.138 | 0.138 | -0.002 | 0.073 | -0.1 | -0.002 | -0.258 | 0.362 |
| Chara | -0.009 | 0.156 | -0.607 | -0.008 | 0.053 | -0.252 | -0.01 | 0.077 | -0.464 |
| others | -0.043 | -0.057 | 1.104 | -0.046 | 0.072 | -1.878 | -0.037 | -0.156 | 3.314 |
| **Women's empowerment** | | | | | | | | | |
| **Attitude towards violence** | | | | | | | | | |
| Low (ref) | | | | | | | | | |
| Medium | 0.026 | 0.374 | 4.469 | | | | | | |
| High | 0.031 | 0.786 | 11.264 | | | | | | |
| **Social independence** | | | | | | | | | |
| Low (ref) | | | | | | | | | |
| Medium | | | | -0.017 | 0.56 | -5.457 | | | |
| High | | | | 0.016 | 0.907 | 8.2 | | | |
| **Decision making** | | | | | | | | | |
| Low (ref) | | | | | | | | | |
| Medium | | | | | | | 0.063 | -0.451 | -16.546 |
| High | | | | | | | 0.15 | 0.434 | 37.737 |

**Table 4. Decomposition of the inequalities in the usage of at least four ANC visits by women's empowerment, 2000–2016 EDHS.**

| Variables | Domains of women's empowerment | | | | | | | | |
|---|---|---|---|---|---|---|---|---|---|
| | Attitude to violence (N = 25,911) | | | Social independence (N = 25,911) | | | Decision making (N = 19,480) | | |
| | Elasticity | CI | % | Elasticity | CI | % | Elasticity | CI | % |
| **Maternal age at pregnancy** | | | | | | | | | |
| > = 20 (ref) | | | | | | | | | |
| < 20 | -0.012 | -0.008 | 0.049 | -0.009 | -0.152 | 0.714 | -0.012 | -0.028 | 0.235 |
| **Place of residence** | | | | | | | | | |
| Urban(ref) | | | | | | | | | |
| Rural | -0.354 | -0.052 | 9.581 | -0.354 | -0.047 | 8.714 | -0.341 | -0.032 | 7.373 |
| **Region** | | | | | | | | | |
| Tigray (ref) | | | | | | | | | |
| Afar | -0.006 | -0.052 | 0.156 | -0.006 | -0.034 | 0.1 | -0.008 | -0.067 | 0.358 |
| Amhara | -0.148 | -0.043 | 3.352 | -0.147 | -0.143 | 10.926 | -0.223 | 0.087 | -13.277 |
| Oromia | -0.159 | -0.009 | 0.714 | -0.157 | 0.012 | -0.982 | -0.267 | -0.005 | 0.982 |
| Somali | -0.021 | 0.027 | -0.302 | -0.021 | 0.048 | -0.525 | -0.026 | -0.202 | 3.647 |
| Benishangul | -0.002 | 0.053 | -0.061 | -0.002 | -0.049 | 0.053 | -0.004 | -0.06 | 0.177 |
| SNNPR | -0.007 | -0.035 | 0.119 | -0.006 | 0.084 | -0.259 | -0.048 | -0.117 | 3.868 |
| Gambella | 0.001 | 0.106 | 0.068 | 0.001 | 0.032 | 0.021 | 0.001 | -0.11 | -0.049 |
| Harar | -0.001 | 0.246 | -0.13 | -0.001 | 0.169 | -0.087 | -0.002 | 0.142 | -0.168 |
| Addis Ababa | 0.033 | 0.547 | 9.396 | 0.033 | 0.47 | 8.182 | 0.031 | 0.251 | 5.357 |
| Dire Dawa | 0.001 | 0.255 | 0.081 | 0.001 | 0.231 | 0.078 | 0.000 | 0.102 | 0.008 |
| **Religion** | | | | | | | | | |
| Orthodox (ref) | | | | | | | | | |
| Protestant | -0.029 | -0.024 | 0.364 | -0.031 | 0.086 | -1.395 | -0.027 | -0.056 | 1.054 |
| Muslim | 0.019 | -0.027 | -0.262 | 0.018 | -0.022 | -0.214 | 0.008 | -0.054 | -0.28 |
| Traditional/other | -0.013 | -0.069 | 0.462 | -0.013 | 0.003 | -0.022 | -0.014 | -0.151 | 1.486 |
| **Media exposure** | | | | | | | | | |
| No exposure (ref) | | | | | | | | | |
| Exposed to 1 media | 0.014 | 0.141 | 1.035 | 0.014 | 0.115 | 0.857 | 0.018 | 0.073 | 0.885 |
| Exposed to 2 media | 0.006 | 0.422 | 1.379 | 0.006 | 0.456 | 1.412 | 0.007 | 0.207 | 1.063 |
| Exposed to 3 media | 0.006 | 0.597 | 1.983 | 0.006 | 0.717 | 2.373 | 0.007 | 0.293 | 1.456 |
| **Wealth** | | | | | | | | | |
| Poorest (ref) | | | | | | | | | |
| Poorer | 0.045 | -0.08 | -1.873 | 0.045 | -0.066 | -1.551 | 0.059 | -0.045 | -1.804 |
| Middle | 0.064 | -0.057 | -1.881 | 0.064 | -0.054 | -1.799 | 0.078 | -0.036 | -1.94 |
| Richer | 0.088 | -0.025 | -1.137 | 0.089 | -0.024 | -1.102 | 0.109 | 0.023 | 1.709 |
| Richest | 0.195 | 0.239 | 24.261 | 0.197 | 0.218 | 22.35 | 0.216 | 0.147 | 21.778 |
| **Women occupation** | | | | | | | | | |
| No occupation (ref) | | | | | | | | | |
| Has occupation | 0.059 | 0.015 | 0.452 | 0.058 | 0.008 | 0.244 | 0.069 | 0.073 | 3.444 |
| **Partner occupation** | | | | | | | | | |
| No occupation (ref) | | | | | | | | | |
| Has occupation | -0.092 | -0.003 | 0.128 | -0.082 | -0.002 | 0.082 | -0.074 | -0.006 | 0.293 |
| **Year of survey** | | | | | | | | | |
| 2000 (ref) | | | | | | | | | |
| 2005 | 0.038 | -0.102 | -1.999 | 0.039 | -0.035 | -0.71 | ref | | |
| 2011 | 0.059 | 0.066 | 2.023 | 0.062 | 0.027 | 0.879 | 0.018 | -0.021 | -0.266 |
| 2016 | 0.218 | 0.142 | 16.137 | 0.221 | 0.066 | 7.606 | 0.186 | 0.124 | 15.869 |

(*Continued*)

**Table 4.** (Continued)

| Variables | Domains of women's empowerment | | | | | | | | |
|---|---|---|---|---|---|---|---|---|---|
| | Attitude to violence (N = 25,911) | | | Social independence (N = 25,911) | | | Decision making (N = 19,480) | | |
| | Elasticity | CI | % | Elasticity | CI | % | Elasticity | CI | % |
| **Maternal education** | | | | | | | | | |
| No education (ref) | | | | | | | | | |
| Primary | 0.076 | 0.078 | 3.103 | 0.072 | 0.14 | 5.257 | 0.075 | 0.032 | 1.618 |
| Secondary | 0.055 | 0.379 | 10.814 | 0.049 | 0.618 | 15.651 | 0.063 | 0.228 | 9.801 |
| Higher | 0.012 | 0.64 | 3.939 | 0.009 | 0.836 | 3.818 | 0.014 | 0.322 | 3.145 |
| **Partner education** | | | | | | | | | |
| No education(ref) | | | | | | | | | |
| Primary | 0.052 | 0.013 | 0.358 | 0.054 | 0.002 | 0.056 | 0.051 | -0.002 | -0.068 |
| Secondary | 0.031 | 0.186 | 2.988 | 0.032 | 0.203 | 3.382 | 0.033 | 0.095 | 2.169 |
| Higher | 0.014 | 0.488 | 3.579 | 0.015 | 0.524 | 4.043 | 0.013 | 0.235 | 2.161 |
| **Ethnicity** | | | | | | | | | |
| Tigrie (ref) | | | | | | | | | |
| Amhara | -0.002 | 0.165 | -0.139 | -0.001 | 0.032 | -0.019 | 0.014 | 0.189 | 1.83 |
| Oromo | -0.054 | 0.081 | -2.268 | -0.054 | 0.039 | -1.108 | -0.038 | 0.114 | -2.94 |
| Afar | -0.038 | -0.113 | 2.221 | -0.038 | -0.138 | 2.698 | -0.005 | 0.037 | -0.138 |
| Somali | -0.019 | 0.129 | -1.282 | -0.019 | -0.056 | 0.544 | -0.003 | 0.043 | -0.095 |
| Guragie | 0.000 | 0.35 | -0.016 | 0.000 | 0.162 | 0.001 | 0.002 | 0.073 | 0.083 |
| Sidama | -0.008 | 0.122 | -0.502 | -0.008 | -0.018 | 0.07 | -0.007 | -0.035 | 0.166 |
| Wolaita | -0.003 | 0.126 | -0.227 | -0.003 | 0.166 | -0.3 | -0.002 | 0.093 | -0.154 |
| Berta | 0.001 | 0.037 | 0.013 | 0.001 | -0.036 | -0.013 | 0.001 | -0.141 | -0.102 |
| Anywak | 0.000 | 0.228 | -0.023 | 0.000 | 0.111 | -0.011 | 0.000 | 0.075 | -0.009 |
| Donga | 0.009 | -0.02 | -0.095 | 0.009 | 0.116 | 0.545 | 0.017 | -0.143 | -1.621 |
| Ari | 0.001 | 0.242 | 0.172 | 0.001 | 0.254 | 0.17 | 0.003 | 0.057 | 0.114 |
| Mossiye | -0.022 | -0.112 | 1.276 | -0.022 | -0.025 | 0.287 | -0.009 | -0.128 | 0.754 |
| Benchi | -0.001 | 0.063 | -0.039 | -0.001 | -0.008 | 0.005 | -0.001 | 0.031 | -0.023 |
| Bodi | 0 | 0.081 | 0.02 | 0.001 | 0.05 | 0.013 | 0.018 | 0 | 0.001 |
| Qewama | -0.007 | -0.009 | 0.032 | -0.006 | -0.079 | 0.262 | -0.011 | 0.017 | -0.13 |
| Burji | -0.002 | -0.007 | 0.007 | -0.002 | 0.063 | -0.059 | 0 | 0.014 | -0.004 |
| Bena | -0.003 | -0.138 | 0.184 | -0.003 | 0.073 | -0.101 | -0.002 | -0.258 | 0.294 |
| Chara | -0.003 | 0.156 | -0.24 | -0.003 | 0.053 | -0.08 | -0.004 | 0.077 | -0.203 |
| others | -0.11 | -0.057 | 3.25 | -0.111 | 0.072 | -4.192 | -0.093 | -0.156 | 9.989 |
| **Women's empowerment** | | | | | | | | | |
| **Attitude towards violence** | | | | | | | | | |
| Low (ref) | | | | | | | | | |
| Medium | 0.005 | 0.374 | 1.068 | | | | | | |
| High | 0.021 | 0.786 | 8.557 | | | | | | |
| **Social independence** | | | | | | | | | |
| Low (ref) | | | | | | | | | |
| Medium | | | | 0.006 | 0.56 | 1.849 | | | |
| High | | | | 0.016 | 0.907 | 7.591 | | | |
| **Decision making** | | | | | | | | | |
| Low (ref) | | | | | | | | | |
| Medium | | | | | | | 0.048 | -0.451 | -14.926 |
| High | | | | | | | 0.123 | 0.434 | 36.524 |

**Table 5. Decomposition of the inequalities in the usage of PNC within 2 days of birth by women's empowerment, 2000–2016 EDHS.**

| Variables | Domains of women's empowerment | | | | | | | | |
|---|---|---|---|---|---|---|---|---|---|
| | Attitude to violence (N = 14,652) | | | Social independence (N = 14,652) | | | Decision making (N = 11,312) | | |
| | Elasticity | CI | % | Elasticity | CI | % | Elasticity | CI | % |
| **Maternal age at birth** | | | | | | | | | |
| > = 20 (ref) | | | | | | | | | |
| < 20 | 0.006 | -0.019 | -0.050 | 0.008 | -0.195 | -0.589 | 0.004 | -0.03 | -0.055 |
| **Place of residence** | | | | | | | | | |
| Urban(ref) | | | | | | | | | |
| Rural | -0.376 | -0.052 | 8.058 | -0.371 | -0.047 | 6.787 | -0.489 | -0.032 | 7.43 |
| **Region** | | | | | | | | | |
| Tigray (ref) | | | | | | | | | |
| Afar | -0.006 | -0.052 | 0.12 | -0.006 | -0.034 | 0.073 | -0.01 | -0.067 | 0.308 |
| Amhara | -0.152 | -0.043 | 2.724 | -0.152 | -0.143 | 8.361 | -0.231 | 0.087 | -9.661 |
| Oromia | -0.325 | -0.009 | 1.158 | -0.327 | 0.012 | -1.516 | -0.483 | -0.005 | 1.248 |
| Somali | -0.007 | 0.027 | -0.084 | -0.008 | 0.048 | -0.141 | -0.014 | -0.202 | 1.366 |
| Benishangul | -0.006 | 0.053 | -0.123 | -0.006 | -0.049 | 0.107 | -0.01 | -0.06 | 0.276 |
| SNNPR | -0.058 | -0.035 | 0.832 | -0.059 | 0.084 | -1.909 | -0.123 | -0.117 | 6.926 |
| Gambella | 0.000 | 0.106 | 0.002 | 0.000 | 0.032 | 0.001 | 0.000 | -0.11 | 0.023 |
| Harar | 0.000 | 0.246 | 0.004 | 0.000 | 0.169 | 0.001 | 0.000 | 0.142 | -0.028 |
| Addis Ababa | -0.013 | 0.547 | -2.82 | -0.013 | 0.47 | -2.317 | -0.028 | 0.251 | -3.377 |
| Dire Dawa | -0.001 | 0.255 | -0.152 | -0.001 | 0.231 | -0.134 | -0.003 | 0.102 | -0.141 |
| **Religion** | | | | | | | | | |
| Orthodox (ref) | | | | | | | | | |
| Protestant | 0.005 | -0.024 | -0.047 | 0.006 | 0.086 | 0.192 | 0.02 | -0.056 | -0.53 |
| Muslim | -0.056 | -0.027 | 0.621 | -0.055 | -0.022 | 0.476 | -0.071 | -0.054 | 1.846 |
| Traditional/other | -0.021 | -0.069 | 0.608 | -0.021 | 0.003 | -0.027 | -0.027 | -0.151 | 1.992 |
| **Media exposure** | | | | | | | | | |
| No exposure (ref) | | | | | | | | | |
| Exposed to 1 media | 0.031 | 0.141 | 1.794 | 0.031 | 0.115 | 1.358 | 0.035 | 0.073 | 1.222 |
| Exposed to 2 media | 0.018 | 0.422 | 3.067 | 0.017 | 0.456 | 3.029 | 0.023 | 0.207 | 2.263 |
| Exposed to 3 media | 0.006 | 0.597 | 1.586 | 0.007 | 0.717 | 1.805 | 0.008 | 0.293 | 1.131 |
| **Wealth** | | | | | | | | | |
| Poorest (ref) | | | | | | | | | |
| Poorer | 0.034 | -0.08 | -1.123 | 0.034 | -0.066 | -0.871 | 0.047 | -0.045 | -1.01 |
| Middle | 0.05 | -0.057 | -1.166 | 0.05 | -0.054 | -1.044 | 0.077 | -0.036 | -1.337 |
| Richer | 0.074 | -0.025 | -0.757 | 0.074 | -0.024 | -0.678 | 0.098 | 0.023 | 1.084 |
| Richest | 0.239 | 0.239 | 23.553 | 0.24 | 0.218 | 20.138 | 0.283 | 0.147 | 20.116 |
| **Women occupation** | | | | | | | | | |
| No occupation (ref) | | | | | | | | | |
| Has occupation | 0.027 | 0.015 | 0.165 | 0.027 | 0.008 | 0.084 | 0.017 | 0.073 | 0.603 |
| **Partner occupation** | | | | | | | | | |
| No occupation (ref) | | | | | | | | | |
| Has occupation | -0.194 | -0.006 | 0.491 | -0.191 | -0.006 | 0.421 | -0.199 | -0.009 | 0.874 |
| **Year of survey** | | | | | | | | | |
| 2000 (ref) | | | | | | | | | |
| 2005 | 0.104 | -0.102 | -4.349 | 0.104 | -0.035 | -1.429 | ref | | |
| 2011 | 0.115 | 0.066 | 3.099 | 0.116 | 0.027 | 1.222 | 0.085 | -0.021 | -0.875 |
| 2016 | 0.328 | 0.142 | 19.242 | 0.328 | 0.066 | 8.377 | 0.343 | 0.124 | 20.547 |

(*Continued*)

**Table 5.** (Continued)

| Variables | Domains of women's empowerment | | | | | | | | |
|---|---|---|---|---|---|---|---|---|---|
| | Attitude to violence (N = 14,652) | | | Social independence (N = 14,652) | | | Decision making (N = 11,312) | | |
| | Elasticity | CI | % | Elasticity | CI | % | Elasticity | CI | % |
| **Maternal education** | | | | | | | | | |
| No education (ref) | | | | | | | | | |
| Primary | 0.09 | 0.078 | 2.896 | 0.084 | 0.14 | 4.526 | 0.103 | 0.032 | 1.57 |
| Secondary | 0.057 | 0.379 | 8.858 | 0.052 | 0.618 | 12.516 | 0.069 | 0.228 | 7.539 |
| Higher | 0.024 | 0.64 | 6.446 | 0.023 | 0.836 | 7.404 | 0.029 | 0.322 | 4.56 |
| **Partner education** | | | | | | | | | |
| No education(ref) | | | | | | | | | |
| Primary | 0.02 | 0.013 | 0.111 | 0.021 | 0.002 | 0.016 | 0.02 | -0.002 | -0.019 |
| Secondary | 0.018 | 0.186 | 1.379 | 0.019 | 0.203 | 1.475 | 0.022 | 0.095 | 1.031 |
| Higher | 0.01 | 0.488 | 2.026 | 0.01 | 0.524 | 2.111 | 0.012 | 0.235 | 1.331 |
| **Ethnicity** | | | | | | | | | |
| Tigrie (ref) | | | | | | | | | |
| Amhara | -0.02 | 0.165 | -1.369 | -0.02 | 0.032 | -0.247 | -0.012 | 0.189 | -1.101 |
| Oromo | -0.033 | 0.081 | -1.083 | -0.032 | 0.039 | -0.488 | -0.021 | 0.114 | -1.144 |
| Afar | -0.07 | -0.113 | 3.255 | -0.068 | -0.138 | 3.64 | -0.01 | 0.037 | -0.172 |
| Somali | -0.028 | 0.129 | -1.458 | -0.028 | -0.056 | 0.594 | -0.019 | 0.043 | -0.39 |
| Guragie | -0.001 | 0.35 | -0.164 | -0.001 | 0.162 | -0.075 | 0 | 0.073 | 0.005 |
| Sidama | -0.004 | 0.122 | -0.219 | -0.004 | -0.018 | 0.029 | -0.003 | -0.035 | 0.057 |
| Wolaita | -0.01 | 0.126 | -0.511 | -0.01 | 0.166 | -0.634 | -0.012 | 0.093 | -0.527 |
| Berta | 0.000 | 0.037 | 0.005 | 0.000 | -0.036 | -0.005 | 0.001 | -0.141 | -0.091 |
| Anywak | 0.000 | 0.228 | -0.032 | 0.000 | 0.111 | -0.014 | 0 | 0.075 | -0.014 |
| Donga | -0.013 | -0.02 | 0.103 | -0.013 | 0.116 | -0.571 | 0.002 | -0.143 | -0.113 |
| Ari | 0.001 | 0.242 | 0.079 | 0.001 | 0.254 | 0.081 | 0.003 | 0.057 | 0.072 |
| Mossiye | -0.039 | -0.112 | 1.788 | -0.038 | -0.025 | 0.361 | 0.037 | -0.128 | -2.302 |
| Benchi | -0.001 | 0.063 | -0.013 | -0.001 | -0.008 | 0.002 | -0.001 | 0.031 | -0.017 |
| Bodi | -0.012 | 0.081 | -0.397 | -0.012 | 0.05 | -0.228 | 0.001 | 0.000 | 0.000 |
| Qewama | -0.025 | -0.009 | 0.092 | -0.025 | -0.079 | 0.757 | -0.018 | 0.017 | -0.148 |
| Burji | 0.00 | -0.007 | -0.002 | 0.001 | 0.063 | 0.013 | 0.003 | 0.014 | 0.017 |
| Bena | -0.004 | -0.138 | 0.234 | -0.004 | 0.073 | -0.115 | -0.004 | -0.258 | 0.463 |
| Chara | -0.008 | 0.156 | -0.543 | -0.009 | 0.053 | -0.175 | -0.012 | 0.077 | -0.461 |
| others | -0.177 | -0.057 | 4.125 | -0.177 | 0.072 | -4.935 | -0.179 | -0.156 | 13.482 |
| **Women's empowerment** | | | | | | | | | |
| **Attitude towards violence** | | | | | | | | | |
| Low (ref) | | | | | | | | | |
| Medium | -0.018 | 0.374 | -2.734 | | | | | | |
| High | -0.004 | 0.786 | -1.291 | | | | | | |
| **Social independence** | | | | | | | | | |
| Low (ref) | | | | | | | | | |
| Medium | | | | 0.016 | 0.56 | 3.433 | | | |
| High | | | | 0.005 | 0.907 | 1.792 | | | |
| **Decision making** | | | | | | | | | |
| Low (ref) | | | | | | | | | |
| Medium | | | | | | | 0.081 | -0.451 | -17.577 |
| High | | | | | | | 0.109 | 0.434 | 22.846 |

To our knowledge, there are no studies addressing women's empowerment inequalities in the use of maternal health care services using the recently developed women's empowerment indices [30]. Direct comparison of our findings with those of other studies is complicated by significant discrepancies in how women's empowerment is measured. However, our findings broadly agree with those of previous research, despite differences in how women's empowerment is measured between our study and that of the literature, that women's empowerment raises the uptake of maternal health care services [37]. The empowerment of women generally tends to enhance the rate at which maternity and child health care services are used, according to a comprehensive analysis of studies from the developing world [17].

The majority of women's empowerment-based disparities in service usage are rooted in the places where women live, the educational levels of women and their partners, the wealth status of households, the period of time the study was conducted, the domains of women's empowerment themselves, mothers' religious affiliation, and media exposure. These findings aid in the development of policies that specifically address the systematic bias that led to differential accumulations of social determinants between women who are empowered and not empowered. The richest category of wealth, for instance, is disproportionately composed of highly empowered women. Given that greater affluence has a significant impact on health care service use in and of itself [38, 39], women from the wealthiest backgrounds face a higher chance of using the service than do their counterparts, therefore widening the gap between women with high levels of empowerment and those with low levels, in terms of access to maternal health care services.

In Ethiopia, government health care spending per capita increased over the last two decades [40] and the overall coverage of the maternal health care services has similarly improved [23, 26]. However, the sheer rise in health care expenditure may not automatically guarantee health care service equity and may even create inequalities in the utilization of maternal health care services as economically richer people are likely to take more of the services than do the poor [41]. This suggests that only highly empowered women are given the privilege of receiving the services because better wealth is more concentrated among these groups of women (richest wealth has positive concentration index, see Tables 2–5). Our results are in line with data from other parts of Ethiopia that show that over time, gains in women's empowerment had been heavily concentrated among the wealthiest wealth [16]. Women's wealth needs to be significantly increased to get them to a point where they can use health care services without facing financial troubles, in order to remove the influence on inequality of wealth's unjust allocation between highly empowered and poorly empowered women. Adopting redistributive measures seeking to evenly divide wealth between women who are now classified as highly empowered and poorly empowered will help to solve an "inverse care law" dilemma [41].

Education, mainly higher education of women, impacts the health care services equity in much the same way that wealth does. Better education highly concentrates among women who are empowered better and functions by widening the inequality in the receipt of the services. The inequality in education between empowered and poorly empowered women happens to lead to inequalities in receiving the services between them, to the disadvantage of the latter group. The positive association between education and women's empowerment is due partly to the fact that education status was part of the variables used to construct the empowerment index. Again, inequalities in the use of the maternal health care services could be narrowed through redistributive policies aiming to equitably distribute education services between empowered and poorly empowered groups of women [41].

Place of residence assumed a completely different mechanisms to affect the disparities of maternal health care services use across women's empowerment categories. Social determinants of health with negative concentration index and elasticity but positive percentage

contribution (see Tables 2–5 in the result section) like rural residence, help widen the unequal utilization distribution of maternal health care services between empowered and poorly empowered women by preventing the latter to not get the services. For example, compared to living in urban areas, rural living has negative elasticities and concentration index and associated with the decrease in the utilization of the services. Since most of the less empowered women live in rural areas, the finding imply that it is the low empowered women who receive far lower proportion of the services, ending up causing the service use gap to grow larger. Since the poorly empowered women cluster in rural areas and in regions such as Afar, Amhara, and SNNPR (variables with negative concentration index, see Tables 2–5), women who reside in these places need to be highly prioritized to benefit them from women's empowerment programs, without ignoring women living in other places. The results of our study are consistent with those of earlier studies, which found that disparities in women's empowerment between rural and urban areas and across Ethiopian regions were pervasive [16].

The finding that women's empowerment-based inequalities in the utilization of maternal health care services underpinned by unequal distributions of social determinants of health can help policymakers address the root causes. However, women's empowerment itself, as an independent exposure variable in its own right, is also responsible for the observed inequalities though its contribution varied by the maternal health care services. For example, for the inequality in the receipt of at least four ANC service, decision making has 15% and 37% contribution respectively for the medium and high categories.

The study's findings pertain to how important it is to empower all women in order to avert women's empowerment related inequalities in maternal health care services. Gender equality and women's empowerment are prioritized in the 2030 Sustainable Development Goals (SDGs) [42], and achieving this goal is essential to achieving all 17 of the SDGs, including the goal on universal access to reproductive health care services [43]. However, gender inequality has persisted, violence against women has even worsened in many parts of the world, and women's health services have been degraded [44]. Despite the fact that COVID-19 pandemic is to blame for the worsening situations surrounding sexual and reproductive health globally, it is also clear that women's health care services were poorly funded already and governments must do more to eliminate systemic drivers of gender inequalities such as gaps in policies, laws and institutions [44].

Even though Ethiopia has made some progress since 2005, empowering women still faces significant obstacles [16], and further funding is needed if the SDG 5 is to be achieved by 2030. A ten-year development plan, covering the years 2021–30, was created by the government. Despite the fact that the plan mentions gender and social inclusion [45], gender equality and women's empowerment don't receive the attention they deserve in the strategic plan. Disparities in the utilization of maternal health services in general, as well as in connection with women's empowerment, have gone unaddressed [45]. It is unlikely that women would be properly empowered socially, economically, legally, psychologically and politically with this document's weak gender reflection, which would have the repercussions of persisting inequities in the provision of maternal health care services.

Equity in maternal health care services also necessitates making the assumption that women are heterogeneous groups and that they should be further segmented by their characteristics such as education, wealth, residence, and age. Researchers and policymakers can now better understand how disparities in maternal health result from the simultaneous action of many identities that either oppress or benefit women thanks to the intersectionality theory [46]. Women who are poor, uneducated, and reside in rural areas, for instance, are distinct from and less likely to use services than women who are wealthy, educated, and reside in urban areas [47]. The theory is beneficial because it prevents us from ignoring the particular

difficulties experienced by women- in accessing the health services- who fall into the first category and are members of disproportionately affected groups [47].

The EDHSs are nationally representative studies and the findings from them can be applied to the population of Ethiopia. The samples were made from the different contexts and locations in the country. Generally, the exposure variables are defined to be comparable across studies and the outcome variables were defined based on accepted standards, which would increase the applicability of the findings to other settings similar in context to Ethiopia.

The study has a few strengths. First, the study used a validated and comparable index for measuring women's empowerment. Unlike studies in the literature, which measured empowerment of women differently, our study employed the new index and would facilitate comparability of findings between studies. Many studies on the same area were done with the use of the standard CI for bounded variable and would likely result in biased findings. The use of the Erreygers normalized concentration index in our study would help us assess the disparities correctly. Also, since the study was done on the pooled data, it increased the sample size and power of the study.

However, the study is not without drawbacks. Interpretations of the findings need to not be made as if the study presents the causes of the observed inequalities; it showed associations. However, some variables' temporality is clearly known (for example, living in urban or rural areas comes first before a woman' use or not use of the ANC) and would suggest possible influence of these variables on access to ANC. Moreover, survey based studies like ours could at least be used as an input for future meta-analysis studies since no single study, even if it is randomized control trial, is generally conclusive and meta-analytic studies have thus been suggested to increase credibility of studies' findings [48]. The decision-making domain of empowerment was not calculated for the 2000 EDHS as some variables used to create it were not measured, and thus decision-making related inequalities were measured based on data from the other three EDHSs.

Furthermore, while the women's empowerment index used in the study helps researchers produce globally comparable studies, at least across LMICs contexts, this index is not without blemish. As some of its domains, such legal, political, and psychological are absent, it falls short of expressing the complete spectrum of all facets of women's empowerment.

## Conclusions

This research reveals that maternal health care services are being used disproportionately by different women's empowerment groups, with highly empowered women using the majority of the services. Unequal distribution of wealth, education, place of residence, and women's empowerment indices itself are some of the key drivers to the unequal consumption of the services.

Women's empowerment-based inequalities of maternal health care services can be improved in different ways. First, national and regional policies must put gender equality and women's empowerment front and center. Every woman should be able to exercise her right to empowerment. Higher level government papers, like the ten-year strategic plan, could think about integrating the intersectional theory framework to help with data collecting and analysis broken down by women's attributes, including wealth and education, to capture double discrimination. It's also important to address differences in how social determinants of health are distributed across women with high levels of empowerment and those with low levels of empowerment. Pro-poor and redistributive policies must receive a lot of attention because weakly empowered women are more likely to also lack wealth, education, and other socioeconomic determinants.

## Supporting information

**S1 File. Stata lines of codes used in the article to prepare and use the design elements.**
(DOCX)

**S1 Checklist. STROBE statement—Checklist of items that should be included in reports of observational studies.**
(DOCX)

## Acknowledgments

We owe the DHS program for granting us access to the data.

## Author Contributions

**Conceptualization:** Gebretsadik Shibre.

**Formal analysis:** Gebretsadik Shibre, Wubegzier Mekonnen, Damen Haile Mariam.

**Investigation:** Gebretsadik Shibre, Wubegzier Mekonnen, Damen Haile Mariam.

**Methodology:** Gebretsadik Shibre.

**Resources:** Gebretsadik Shibre.

**Software:** Gebretsadik Shibre.

**Supervision:** Wubegzier Mekonnen, Damen Haile Mariam.

**Writing – original draft:** Gebretsadik Shibre.

**Writing – review & editing:** Gebretsadik Shibre, Wubegzier Mekonnen, Damen Haile Mariam.

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
