## [Decision Letter · Decision Letter 0]

27 Mar 2023

PONE-D-23-01180Decomposition analysis of women’s empowerment-based inequalities in the use of maternal health care services in Ethiopia: Evidence from Demographic and Health SurveysPLOS ONE

Dear Dr. shibre,

Thank you for submitting your manuscript to PLOS ONE. After careful consideration, we feel that it has merit but does not fully meet PLOS ONE’s publication criteria as it currently stands. Therefore, we invite you to submit a revised version of the manuscript that addresses the points raised during the review process.

We look forward to receiving your revised manuscript.

Kind regards,

Kannan Navaneetham, PhD

Academic Editor

PLOS ONE

Reviewers' comments:

Reviewer's Responses to Questions

**Comments to the Author**

1. Is the manuscript technically sound, and do the data support the conclusions?

Reviewer #1: Yes

2. Has the statistical analysis been performed appropriately and rigorously? 

Reviewer #1: Yes

3. Have the authors made all data underlying the findings in their manuscript fully available?

Reviewer #1: Yes

4. Is the manuscript presented in an intelligible fashion and written in standard English?

Reviewer #1: Yes

5. Review Comments to the Author

Reviewer #1: The authors present an interesting retrospective analysis and use women's empowerment as an entry point to explore inequalities in maternal health services. The study is very well written and provides important data to readers of the journal PLOS ONE.

A few suggestions are included below:

1) On page 13, According to the instructions, the article's core variable, women's empowerment, was formed based on SWPER GLOBAL. Still, its formation as an important variable needs to be clearly described. If PCA began it, please provide data for each load to illustrate the applicability of the empowerment scale in Ethiopia. If the calculation was done directly using the code supplied by SWPER GLOBAL, it should also be stated that no PCA was performed but that the composite index method was used for direct formation.

2) On page 30， Education is used as an explanatory variable and a dimension constituting social independence. Education’s prediction of the social independence component may not be scientifically valid, by which I mean, is there literature to support you in doing so?

3) In Table 1, one of the brackets for the Attitude towards violence variable is incorrectly used.

6. PLOS authors have the option to publish the peer review history of their article (what does this mean?). If published, this will include your full peer review and any attached files.

Reviewer #1: No

While revising your submission, please upload your figure files to the Preflight Analysis and Conversion Engine (PACE) digital diagnostic tool, https://pacev2.apexcovantage.com/. PACE helps ensure that figures meet PLOS requirements. To use PACE, you must first register as a user. Registration is free. Then, login and navigate to the UPLOAD tab, where you will find detailed instructions on how to use the tool. If you encounter any issues or have any questions when using PACE, please email PLOS at figures@plos.org. Please note that Supporting Information files do not need this step.<quillbot-extension-portal></quillbot-extension-portal>

---

## [Author Response · Author response to Decision Letter 0]

10 Apr 2023

Response: Thank you. We revised and prepared the materials in accordance with PLOSE ONE’s style requirement. 

2. We note that you have indicated that data from this study are available upon request. PLOS only allows data to be available upon request if there are legal or ethical restrictions on sharing data publicly. 

Response: Thank you. We have clarified the ambiguity of the statement on the availability of the data we based our study on. We used the EDHS data which are made public by the DHS program. There is no ethical or legal restriction to accessing the dataset. The “upon request” phrase has now been omitted. 

Response: Thank you. As we have said above, the EDHS data are available to the public domain and access to it is unrestricted. There are no data we owned privately. We have shown this in our revised cover letter. 

Response: Thank you. I updated my existing ORCID iD. 

Response: Thank you. We have now added ethics statement in the method section. 

Response: Thank you. The references are correct. 

Reviewers' comments:

Reviewer's Responses to Questions

Comments to the Author

1. Is the manuscript technically sound, and do the data support the conclusions?

Reviewer #1: Yes

Response: Thank you.

2. Has the statistical analysis been performed appropriately and rigorously?

Reviewer #1: Yes

Response: Thank you.

3. Have the authors made all data underlying the findings in their manuscript fully available?

Reviewer #1: Yes

Response: Thank you. 

4. Is the manuscript presented in an intelligible fashion and written in standard English?

Reviewer #1: Yes

Response: Thank you.________________________________________

5. Review Comments to the Author

Reviewer #1: The authors present an interesting retrospective analysis and use women's empowerment as an entry point to explore inequalities in maternal health services. The study is very well written and provides important data to readers of the journal PLOS ONE.

Response: Thank you.

A few suggestions are included below:

1) On page 13, According to the instructions, the article's core variable, women's empowerment, was formed based on SWPER GLOBAL. Still, its formation as an important variable needs to be clearly described. If PCA began it, please provide data for each load to illustrate the applicability of the empowerment scale in Ethiopia. If the calculation was done directly using the code supplied by SWPER GLOBAL, it should also be stated that no PCA was performed but that the composite index method was used for direct formation.

Response: Thank you for pointing this issue out. What we wanted to portray was that the authors of the SWPER Global themselves used PCA to drive scores of each variable. By PCA, we only mean that they used it as part of their formation of the composite indices. We simply utilized the codes supplied by them to create these indices in our dataset. We have now clarified this point in the revised paper. 

2) On page 30，Education is used as an explanatory variable and a dimension constituting social independence. Education’s prediction of the social independence component may not be scientifically valid, by which I mean, is there literature to support you in doing so?

Response: We want to thank you again for your comment. Education is not a predictor of social independence but maternal health services. We took into account in our model maternal education because we believed that it could potentially muddle the empowerment-maternal service usage connection. The rationale is that education is one of the many elements that make up the social independence domain. The SWPER Global authors recognize that education can affect women’s empowerment and that is why education has become one of the constituting components of women empowerment, mainly the social independence dimension. A woman has a greater likelihood of achieving social empowerment the more educated she is. So, education and social independence have a link between each other.

Social independence refers to women's empowerment or capacity to achieve their objectives. We also made an effort to clarify what the other domains pertain to so that readers could readily comprehend them, using your insightful suggestion about the need to elaborate more on social independence. Therefore, the nexus between maternal education and women empowerment (social independence) is just association, and in this study, this association was known by looking at the concentration index of education with respect to the ranking variable of social independence. A +ve concentration index of education (especially the higher educational status) means that more of the higher education is concentrated among women who are more empowered. 

3) In Table 1, one of the brackets for the Attitude towards violence variable is incorrectly used. 

Response: Thank you. We have now corrected it.

---

## [Editor Report · Decision Letter 1]

14 Apr 2023

Decomposition analysis of women’s empowerment-based inequalities in the use of maternal health care services in Ethiopia: Evidence from Demographic and Health Surveys

PONE-D-23-01180R1

Dear Dr. shibre,

We’re pleased to inform you that your manuscript has been judged scientifically suitable for publication and will be formally accepted for publication once it meets all outstanding technical requirements.

Kind regards,

Kannan Navaneetham, PhD

Academic Editor

PLOS ONE

Additional Editor Comments (optional):

Reviewers' comments:

<quillbot-extension-portal></quillbot-extension-portal>

---

## [Editor Report · Acceptance letter]

19 Apr 2023

PONE-D-23-01180R1 

Decomposition analysis of women’s empowerment-based inequalities in the use of maternal health care services in Ethiopia: Evidence from Demographic and Health Surveys 

Dear Dr. Shibre:

I'm pleased to inform you that your manuscript has been deemed suitable for publication in PLOS ONE. Congratulations! Your manuscript is now with our production department. 

Kind regards, 

on behalf of

Prof. Kannan Navaneetham 

Academic Editor

PLOS ONE